# Immunotherapy in Extensive-Stage Small Cell Lung Cancer

**Rola El Sayed and Normand Blais ***

Centre Hospitalier de Université de Montreal, Montreal, QC H2X 0C1, Canada; rola.el.sayed@umontreal.ca
* Correspondence: normand.blais.med@ssss.gouv.qc.ca

**Abstract:** Small cell lung cancer (SCLC) remains a poorly understood disease with aggressive features, high relapse rates, and significant morbidity as well as mortality, yet persistently limited treatment options. For three decades, the treatment algorithm of SCLC has been stagnant despite multiple attempts to find alternative therapeutic options that could improve responses and increase survival rates. On the other hand, immunotherapy has been a thriving concept that revolutionized treatment options in multiple malignancies, rendering previously untreatable diseases potentially curable. In extensive stage SCLC, immunotherapy significantly altered the course of disease and is now part of the treatment algorithm in the first-line setting. Nevertheless, the important questions that arise are how best to implement immunotherapy, who would benefit the most, and finally, how to enhance responses.

**Keywords:** immunotherapy; small cell lung cancer; checkpoint inhibitors; PD-1; PD-L1; CTLA-4





## 1. Introduction

Lung cancer remains the leading cause of cancer death worldwide [1]. Major milestones have been achieved in non-small cell lung cancer immunotherapy and targeted treatment with significant improvement in responses, as well as survival; yet small cell lung cancer (SCLC) that accounts for 10–20% of all lung cancer cases staggers behind with therapeutic quiescence that strengthens its status as an aggressive malignancy with a five-year survival rate of around 7% [2]. High proliferative index, rapid doubling time, and strong propensity to metastasize all contribute to its dismal prognosis despite active treatment [3].

The veteran's administration lung cancer study group categorizes SCLC into limited or extensive-stage disease according to whether the disease is limited to one hemithorax in a field amenable to radiation therapy (noting that TNM staging can also be used) [4]. Significant differences have been noted in the characteristics of limited versus extensive-stage disease and their response to different therapeutic approaches [5]. For the scope of this review, we will only be discussing extensive-stage SCLC (ES-SCLC).

Despite multiple therapeutic innovations in the field of ES-SCLC, platinum-etoposide (PE) chemotherapeutic protocol maintains its position as the mainstay first-line treatment of SCLC, as SCLC is quite chemotherapy-sensitive in the first-line setting [6]. However, quick emergence of resistance, transient benefit of therapy, and limited efficacy of subsequent lines [7] compel physicians and scientists to seek better treatment options.

It has always been hypothesized that SCLC is an immunologic disease. A strong correlation with cigarette smoking [8] implies a potentially high tumor mutational burden (TMB) as well as high neoantigen diversity [9]. Furthermore, the established occurrence of auto-immune paraneoplastic manifestations in SCLC, such as Lambert–Eaton myasthenic syndrome, sensory neuropathy, limbic encephalitis, and syndrome of inappropriate secretion of antidiuretic hormone, highlight the strong immunogenic feature of SCLC cells [10,11]. Unfortunately, SCLC initial studies have suggested only modest responses to single-agent immunotherapy, and classical predictive biomarkers of immunotherapy were not found to be of benefit for patients with ES-SCLC.

Nevertheless, two important phase III clinical trials, IMPower133 and CASPIAN, have now shown the significant role and benefit of the integration of immune check-point inhibitors in combination with front-line chemotherapy in the therapeutic algorithm of ES-SCLC, finally changing the treatment paradigm in this setting. More studies are needed to understand the difference in outcomes of patients with ES-SCLC receiving immunotherapy as well as find the proper biomarkers that can identify the patient subcategories that can derive the most benefit.

This review elaborates the new understanding of SCLC biology and discusses the use of immunotherapy in ES-SCLC proposing means to optimize benefits in this patient population.

## 2. Subtypes of SCLC

Whole genome sequencing and transcriptional profiling broadened our understanding of SCLC by recognizing specific patterns which may lead to more rationally designed therapies for SCLC (Figure 1) [12–14]. Loss of function mutations of retinoblastoma1 (Rb1) and TP53 are a ubiquitous finding in SCLC. Several other possibly targetable or non-targetable alterations such as MYC amplification, PTEN loss, PI3K activating mutation, and FGFR1 amplification are also common. DNA damage response (DDR) mediators altered in SCLC represent yet another aspect of understanding the landscape of SCLC biology. These include checkpoint kinase 1 (CHK1), ataxia telangiectasia and RAD3-related protein (ATR), ataxia telangiectasia mutated (ATM), aurora kinase (AURK) A or B, and WEE1. Epigenetic related alterations such as inhibition of enhancer of zeste homology 2 (EZH2) and variations in lysine-specific demethylase 1A (LSD1) were also noted to play a role in tumor immunogenicity as well as response to treatment, and they provide promising anti-tumor targets. Nevertheless, although biologically relevant, no actionable mutation has yet proven to be associated with therapeutic benefits in the clinic [14,15].

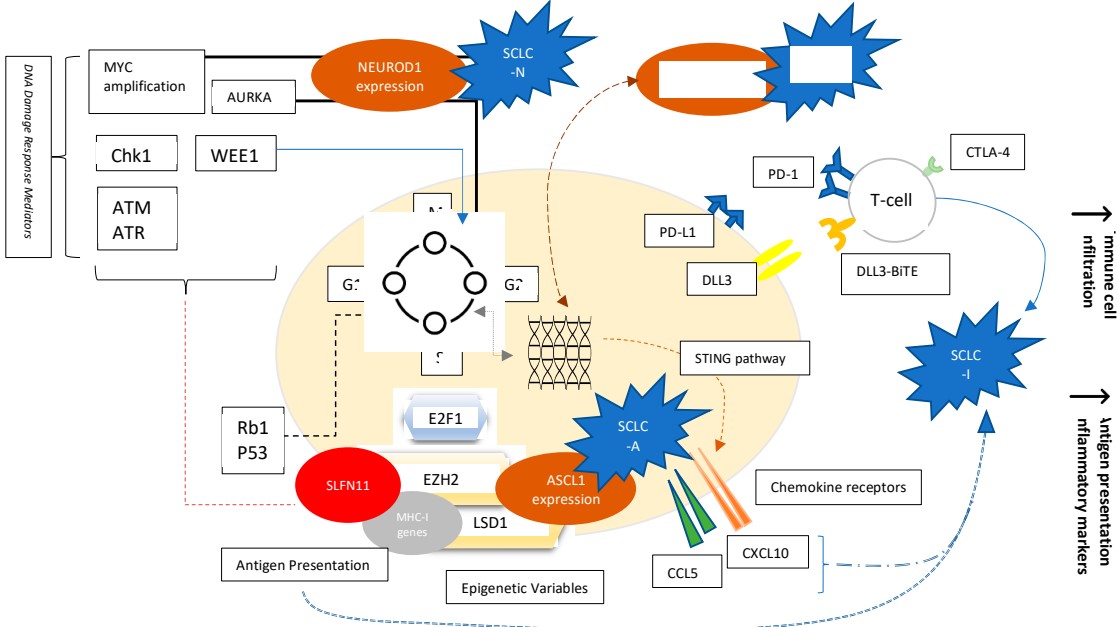

**Figure 1.** Factors affecting SCLC biology. Molecular Variables, Signaling Pathways, Transcription factors, Epigenetics, and Cell-surface Receptors. Evolving concepts in the understanding of SCLC include the following: The ubiquitous loss of TP53 and Rb1 involved mainly in the G1-S phase cellular cycle checkpoints; NEUROD1, ASCL1, and POUF2 expression as determinants of SCLC subtypes; Over-expression of AURKA involved in the G2-M checkpoint, and characterizing the MYC-driven NEUROD1 high SCLC-N subtype; Role of DNA damage response mediators such as Chk1, WEE1, ATM/ATR, as well as AURK; Role of epigenetics such as EZH2 and LSD1; Role of EZH2 influencing response to chemotherapy by alteration of SLFN11 and immune phenotype by effect on MHC-I presentation; Role of EZH2 in affecting ASCL1 expression responsible for SCLC-A phenotype through TGF-beta-SMAD pathway; Variable expression of chemokine receptors dependent on STING pathway; PD-L1 and DLL3 studied as targetable cell-surface receptors.

An important perspective-changing finding in SCLC has been the role of transcription factors in shaping tumor behavior and the characterization of variable biologic subtypes of SCLC. Key transcription factors such as ASCL1, NEUROD1, POU2F3, and YAP1 were found to be responsible for variable neuroendocrine differentiation patterns of SCLC, and investigators were able to recognize four major distinct subcategories using unbiased RNA-sequencing: SCLC-A, SCLC-P, SCLC-N, and finally SCLC-Y that was dropped in more recent classification studies as it was characterized by YAP1 expression, and YAP1 expression has been found to be rather nonspecific to one biologic sub-type. Recent studies have also pointed to the presence of an immune dependent subtype, referred as SCLC-I [16,17]. These subcategories were divided into those with (SCLC-A, SCLC-N) or without (SCLC-P, SCLC-I) neuro-endocrine differentiation. Distinct features and therapeutic vulnerabilities were noted in each sub-category [17,18]. (Table 1)

**Table 1.** SCLC subtypes, key transcription factors, and possible therapeutic approaches.

| SCLC Subtype | Transcription Factor Expression | Possible Targets |
|:---:|:---:|:---:|
| **SCLC-A** | High ASCL1, High DLL3 SLFN11 expression | DLL3 inhibition Platinum-based chemotherapy PARP inhibition BCL-2 inhibition |
| **SCLC-N** | High expression of NEUROD1 High expression of SSTR2 | Somatostatin analogs Aurora Kinase inhibitors |
| **SCLC-P** | High POU2F3 expression | PARP inhibition Anti-metabolites |
| **SCLC-I** | High expression of EMT+ BTK Increased Immune Infiltration (Higher antigen presentation + immune cell infiltration) | Ibrutinib Immune checkpoint inhibitors |

SCLC-A, characterized by high ASCL1 expression, was found to have high delta-like ligand-3 (DLL3) expression. Furthermore, the protein Schlafen 11 (SLFN11), recently found to be a promising predictor of sensitivity to DNA-damaging chemotherapy [19], is most differentially expressed in SCLC-A subtype. This renders SCLC-A subtype potentially more sensitive to DLL-3 inhibition as well more responsive to platinum therapy and PARP inhibition. It was also noted that SCLC-A has an increased susceptibility to BCL2 inhibition [17].

SCLC-N, on the other hand, is characterized by high expression of NEUROD1. Early observational data showed over-presentation of SCLC-N variant in recurrent, chemo-resistant disease, suggesting cell lines to be derived from previously treated individuals, probably related to tumor evolution [16]. Notably, SCLC-N phenotypes seemed to favor epithelial mesenchymal transformation (known to associated with distant metastasis), and axonogenesis, possibly implying an increased propensity to central nervous system metastases [20]. It also highly expresses the surface protein somatostatin receptor 2 (SSTR2), which can be targetable by somatostatin analogs. Moreover, SCLC-N tumors were noted to have increased sensitivity to aurora kinase inhibitors [17].

SCLC-P, accounting for 7% of SCLC cases [21], has high POU2F3 expression. It is associated with lack of DLL3 expression. However, it was found to be particularly vulnerable to PARP inhibition and antimetabolites [17].

SCLC-Y, initially found in 5–10% of SCLC tumors, more commonly in tumors of mixed histology previously characterized with high YAP1 expression, enriched RB wild-type status, and low or absent expression of ASCL1, and NEUROD1 [22], was considered to be of poor prognosis, with a tendency to chemoresistance. It was discussed to be more immunogenic and defined as a distinct T-cell inflamed phenotype [22]. Eventually, it was noted that YAP1 expression was not restricted to a specific phenotype and is usually expressed in rare cases across all SCLC subtypes [21], but rather another SCLC-I subtype

with low expression of transcription factors and an inflamed gene signature could be identified [23].

Finally, SCLC- I, or "inflamed" SCLC neuro-endocrine-low subtype, lacks the expression of ASCL1, NEUROD1, and POU2F3. It exhibits high expression of epithelial-mesenchymal transition and increased immune cell infiltration, enhanced antigen presentation machinery, and more interferon gamma activation suggesting a particular relevance of checkpoint inhibition in this setting. SCLC-I cells express Bruton tyrosine kinase (BTK) in excess as well, rendering them possibly sensitive to inhibition by BTKi such as Ibrutinib [17].

In absence of definitive correlation of response to immunotherapy and conventional biomarkers, perhaps the identification of SCLC-I subtype will help us better predict the effectiveness of ICIs [24], as was discussed by Gay et al. during their retrospective review of IMpower 133 cases regarding SCLC subtypes [17], which will be discussed later in this review. Immunohistochemistry has been attempted for the easier and faster identification of subtypes [25]; however, it remains a gray zone with inadequate characterization.

Interesting enough, when investigating subtype variability, and significant tumor diversity as well as interpatient dynamic tumor heterogeneity that can be responsible for treatment resistance, as well as change into more pro-metastatic features, sample clustering and different signatures of cell line plasticity were noted [21]. Single-cell atlas of human small cell lung cancer assay studying transcriptional heterogeneity demonstrated the role of PLCG2 expression in substantial phenotypic changes in the SCLC immune microenvironment, with PLCG2-high tumors having a more profoundly immuno-suppressed microenvironment and being more prone to metastasis [20]. Further genetic scrutiny in mouse models regarding key features of phenotypic switch in SCLC variants led to the discovery of the role of TAZ or transcriptional coactivator with PDZ-binding motif as an important downstream mediator of SWI/SNF complex responsible for morphologic and phenotypic behavior of SCLC cell lines [26]. The discovery of these possibly targetable genetic/epigenetic alterations responsible for phenotypic switching and consequently therapeutic resistance signals possible means to bypass SCLC treatment resistance leading to potential enhancement of treatment responses via synergistic combinations.

## 3. Role of Immunotherapy in SCLC

SCLC is an aggressive disease with high potential for metastasis, and for more than three decades, treatment of ES-SCLC remained platinum-based chemotherapy, despite high rates of treatment resistance and disease recurrence. Multiple therapeutic approaches have been attempted with no success in the past, until the advent of immunotherapy. Considering the aggressivity of the disease and the expectation that chemotherapy induction is still required for these patients, no study included immunotherapy as a single-agent approach in the first-line setting. Two mature large phase III clinical trials are now published and have led to a long-awaited shift in the treatment paradigm improving disease responses as well as patient outcomes. IMpower 133 [27] and CASPIAN [28] have demonstrated substantial evidence of benefit from adding atezolizumab and durvalumab respectively to chemotherapy, leading to the approval of these check point inhibitors (CPIs) in the frontline setting of ES-SCLC. Nevertheless, the absolute benefit of programmed death ligand 1 (PD-L1) inhibitors in SCLC is more modest than for NSCLC highlighting the need for a better understanding of the SCLC phenotypes and immune signatures that may offer a more rational approach to patient selection.

## 4. Clinical Studies of Immunotherapy in ES-SCLC

*4.1. First Line Studies*

4.1.1. IMpower133

In IMpower133, investigators assessed the safety and efficacy of the addition of atezolizumab, an inhibitor of PD-L1, versus placebo to first-line carboplatin and etoposide in four 21-day cycles, followed by a maintenance of atezolizumab versus placebo based on the initial randomization, in a 1:1 double-blind randomization of 403 treatment-naive ES-

SCLC. The study was positive, showing an improved overall response rate (ORR) of 64.4% versus 60.2% in favor of atezolizumab. In addition, there was a statistically significant improvement of median progression free survival (mPFS) (5.2 versus 4.3 months in favor of atezolizumab, hazard ration (HR) 0.77, 95% confidence interval (CI) 0.62–0.96, *p* = 0.02), as well as median overall survival (mOS) (12.3 versus 10.3 months in favor of atezolizumab also, HR 0.7, 95% CI 0.54–0.91, *p* = 0.007). At one year, the overall survival was 13% higher in the atezolizumab group compared to placebo (51.7% vs. 38.2%). Adverse events were mostly hematologic and were observed more often in the combination group but considered consistent with previously reported safety profile [27]. Based on IMpower 133 results, atezolizumab in combination with standard chemotherapy received FDA approval on 18 March 2019.

### 4.1.2. CASPIAN Trial

The CASPIAN trial is another phase III, open label trial where durvalumab, also an anti-PD-L1 inhibitor, was tested in combination with platinum and etoposide (PE, carboplatin or cisplatin allowed), or in addition to tremelimumab, a CTLA-4 inhibitor, (four cycles of induction for each of the two groups, followed by durvalumab maintenance) versus PE alone (six cycles) in 805 treatment-naïve patients with ES-SCLC in a 1:1:1 randomization. Optional prophylactic cranial irradiation (PCI) was allowed after chemotherapy at the investigator's discretion, but not permitted in the immunotherapy groups before discontinuation of all study treatment. Durvalumab plus chemotherapy showed improved responses when compared to PE alone with a mOS of 12.9 versus 10.5 months in favor of durvalumab (HR 0.75, 95% CI 0.62–0.91, *p* = 0.0032), with sustained separation of OS survival curves of 22.2% vs 14.4% in favor of durvalumab at 24 months. The addition of tremelimumab did not significantly improve OS in comparison with EP alone (HR = 0.82; 95% CI 0.68–1; *p* = 0.045); median overall survival was 10.4 months (95% CI 9.6–12.0) versus 10.5 months (9.3–11.2), respectively, with increased risk of adverse events [28,29]. Following the results of CASPIAN, durvalumab became yet another FDA approved option to be given in combination with EP in first line treatment of ES-SCLC on 27 March 2020.

In a recent update of >three-years of follow-up (39.4 months, 86% maturity of data) for censored patients in CASPIAN trial, discussed at the ESMO congress 2021, at the data cut-off date of 27 March 2021, durvalumab addition to chemotherapy showed sustained clinically significant OS benefit in favor of durvalumab with a mOS of 12.9 months versus 10.5 months when compared to chemotherapy alone (HR 0.71, 95% CI 0.6–0.86, *p* = 0.0003). At the two-year landmark, 22.9% versus 13.9% of patients were alive in favor of durvalumab. At 36 months, 17.6% vs. 5.8% of pts were alive also in favor of durvalumab. Regarding the durvalumab+ tremelimumab+ chemotherapy arm, OS was numerically improved versus chemotherapy with 15.3% of patients alive at 36 months, however, without any statistically significant difference (HR 0.81, 95% CI 0.67–0.97; *p* = 0.02) [30].

### 4.1.3. KEYNOTE-604

KEYNOTE-604 is a randomized placebo-controlled double-blind phase III trial where pembrolizumab, a PD-1 inhibitor, versus placebo is combined with EP, followed by pembrolizumab versus placebo maintenance in patients with treatment-naïve ES-SCLC. PFS endpoints were met with an improved mPFS of 4.5 versus 4.3 months in favor of pembrolizumab (HR 0.75, 95% CI 0.65–0.91, *p* = 0.0023). However, despite a numerically improved mOS of 10.8 versus 9.7 months, no statistically significant difference was noted with OS endpoint being missed (HR 0.8, 95% CI 0.64–0.98, *p* = 0.0164).

Interestingly, as in IMpower 133 and CASPIAN, long-term survival favored the pembrolizumab arm. The 12-month PFS was 13.6% with pembrolizumab plus EP and 3.1% with placebo plus EP. Twenty-four-month OS estimates were 22.5% and 11.2%, respectively [31].

### 4.1.4. Arriola et al. (BMS/Cancer Research UK) and Reck et al. (BMS, Phase II)

NCT01331525 (Arriola et al.) is a phase II trial where ipilimumab, a CTLA-4 inhibitor, at a dose of 10 mg/kg was combined with carboplatin and etoposide in chemotherapy naïve ES-SCLC for up to 6 cycles, showing a 72.4% overall response rate (ORR), a median progression free survival (mPFS) of 6.9 months (95% CI 5.5–7.9), and a median overall survival (mOS) of 17 months (95% CI 7.9–24.3). One-year PFS, which was the primary endpoint, reached 15.8%. In this study, five deaths were reported to be related to ipilimumab [32].

In NCT00527735 (Reck et al.), a phase II, randomized double-blind multi-center trial, 130 patients with SCLC were randomized 1:1:1 to receive either concurrent-ipilimumab regimen (four doses of ipilimumab/paclitaxel/carboplatin followed by two doses of placebo/paclitaxel/carboplatin); phased-ipilimumab regimen (two doses of placebo/paclitaxel/carboplatin followed by four doses of ipilimumab/paclitaxel/carboplatin); or the control regimen (up to six doses of placebo/paclitaxel/carboplatin). Carboplatin and paclitaxel with ipilimumab given in a phased manner showed higher immune related best ORR (irBORR) at a rate of 71%, compared to 57% in the concurrent group and 49% in the placebo group. The mPFS ranged at 5.2, 3.9, and 5.2 months, the irPFS was of 6.4, 5.7 and 5.3 months, and the mOS was of 12.9, 9.1, and 9.9 months (HR = 0.75, *p* = 0.13) in the same three groups, respectively. Consequently, a phase III trial was conducted to test the benefit of the phased combination [33].

### 4.1.5. Reck et al. (BMS, Phase III)

In this phase III double-blind trial, NCT01450761, a phased course of ipilimumab versus placebo were combined with etoposide and platinum (carboplatin or cisplatin) in 1132 newly diagnosed ES-SCLC patients. The study showed identical BORR at 62%, with a mPFS of 4.6 months with ipilimumab compared to 4.4 months with placebo (HR 0.85, *p* = 0.016). mOS was 11 vs. 10.9 months with ipilimumab versus placebo (HR 0.94, *p* = 0.38). Moreover, higher rates of toxicity were noted with the combination protocol, with five treatment-related deaths occurring in the ipilimumab arm [34].

### *4.2. Second-Line and beyond Trials*

#### 4.2.1. KEYNOTE-028

KEYNOTE-028 is a phase Ib multi-cohort, open-label trial that investigated the efficacy and safety of pembrolizumab, a PD-1 inhibitor, in recurrent SCLC with PD-L1 positive tumors, showing promising results in patients with pre-treated SCLC, with one patient out of 24 recruited achieving complete response and an ORR reaching 33% (95% CI of 16–55%). Adverse events were acceptable with the most common being asthenia (*n* = 7), fatigue (*n* = 7), and cough (*n* = 6). mPFS was 1.9 months, and mOS was 9.7 months [35].

#### 4.2.2. KEYNOTE-158

KEYNOTE-158 is a larger phase II basket trial using pembrolizumab in 11 cancer types including advanced recurrent SCLC, nevertheless, unlike Keynote-028, regardless of PD-L1 status. Results showed an ORR of 18.7% for the entire population compared to 35.7% for PD-L1 positive tumors, with a mPFS of two months and a mOS of 9.1 months in the entire population, whereas results were of 2.1 and 14.6 months, respectively, in PD-L1 positive patients, suggesting benefit of immunotherapy in the second line and beyond treatment of ES-SCLC. Once again, pembrolizumab seemed to be promising for PD-L1 positive recurrent ES-SCLC [36].

#### 4.2.3. CheckMate 032

CheckMate-032 is a basket phase I/II study, evaluating the activity of nivolumab as monotherapy at a dose of 3 mg/kg or in combination with ipilimumab, a CTLA-4 inhibitor, at doses of nivolumab 1 mg/kg + ipilimumab 3 mg/kg (N1/I3) or nivolumab 3 mg/kg + ipilimumab 1 mg/kg (N3/I1), in several malignancies including refractory ES-SCLC. In platinum-refractory ES-SCLC, the ORR was 10%, mPFS of 1.4 months, and

mOS of 4.4 months with nivolumab as a single agent, compared to an ORR of 23%, mPFS of 2.6 months, and mOS of 7.7 months in the combination group where nivolumab was given at a dose of 1 mg/kg and ipilimumab 3 mg/kg versus an ORR of 19%, mPFS of 1.4 months, and mOS of 6 months in the group where nivolumab was given at a dose of 3 mg/kg and ipilimumab 1 mg/kg. This study suggested that the optimal dose for further studies in this setting was nivolumab at a 1 mg/kg dosing with ipilimumab at a dose of 3 mg/kg. At the data cut-off, the median duration of response (DOR) was not yet reached with single agent nivolumab. In the N3/I1 group, the median DOR was 4.4 months (95% CI, 3.7–not reached). In the N1/I3 group, the median DOR was 7.7 months (95% CI, 4.0–not reached). Adverse events were more common in combination groups.

Preliminary efficacy data combined with an acceptable toxicity profile led to an accelerated FDA approval of nivolumab as a single agent in the salvage setting of SCLC treatment [37].

### 4.2.4. CheckMate-331

CheckMate-331 is also a randomized, open-label, phase III trial comparing nivolumab versus chemotherapy (topotecan or amrubicin), in pre-treated relapsed ES-SCLC, with 284 patients enrolled having received one prior line of cisplatin-based chemotherapy and being stratified into platinum-sensitive and platinum-resistant categories. There was no improvement in OS, with mOS in nivolumab group being 7.5 months compared to 8.4 months in the chemotherapy arm (HR = 0.86, 95% CI 0.72–1.04; $p$ = 0.11), even though a delayed separation of curves was noted after 12 months [38]. Accordingly, FDA approval for nivolumab in ES-SCLC was withdrawn.

### 4.3. Maintenance Trials

### CheckMate-451

CheckMate 451 is a double-blind phase III maintenance trial with 834 patients with ES-SCLC who achieved disease control on first-line platinum-based chemotherapy randomized to receive nivolumab 240 mg every two weeks alone versus nivolumab at 1 mg/kg with ipilimumab 3 mg/kg versus placebo, followed by nivolumab 240 mg once every two weeks or placebo for ≤two years or until progression or unacceptable toxicity. The primary endpoint was OS with nivolumab plus ipilimumab versus placebo. Within a minimal follow-up of 8.9 months, there was no statistically significant difference in nivolumab/ipilimumab versus placebo (9.2 vs. 9.6 months in favor of placebo, HR = 0.92, 95% CI 0.75–1.12, $p$ = 0.37). mOS for nivolumab alone was 10.4 months (HR = 0.84, 95% CI 0.69–1.02 when compared to placebo). PFS for nivolumab/ipilimumab, and nivolumab alone had a HR of 0.72 (95% CI 0.6–0.87) and 0.67 (95% CI 0.56–0.81), respectively. ORR was in favor of the combination subgroup. Moreover, a trend towards OS benefit was noted with nivolumab plus ipilimumab in patients with a higher TMB (>13 mutations per mega-base). Although the primary end point was not met, a tendency towards improved ORR, PFS, and DOR was noted [39].

The field has been and continues to be investigated extensively (Table 2), with several studies ongoing or underway, including immunotherapy alone or in combination with chemotherapy or targeted agents, not only in ES-SCLC, but also in limited stage, in upfront, maintenance, or refractory/recurrent settings.

**Table 2.** Pertinent Studies of Immunotherapy in ES-SCLC.

| Study | Phase | Setting | Agent | Patients | Primary Endpoint | Result |
|-------|-------|---------|-------|----------|------------------|--------|
| **IMpower133** | III | First-line | Atezolizumab | 403 | PFS & OS | Improved PFS and OS in favor of atezolizumab |
| **CASPIAN** | III | First-line | Durvalumab +/− Tremelimumab | 805 | OS | Improved OS in favor of durvalumab. No benefit of addition of tremelimumab. |
| **KEYNOTE-604** | III | First-line | Pembrolizumab | 453 | PFS & OS | Improved PFS in favor of pembrolizumab. No statistically significant difference in OS. |
| **Arriola et al.** | II | First-line | Ipilimumab | 42 | PFS at 1 year | 1-year PFS 15.8% |
| **Reck et al.** | II | First-line | Ipilimumab (+ carboplatin/paclitaxel in a concurrent/ phased manner) | 130 | irPFS | Improved irPFS in favor of Ipilimumab combination in a phased manner |
| **Reck et al.** | III | First-line | Ipilimumab (phased) | 1132 | OS | No difference in survival $p = 0.38$ |
| **CheckMate-451** | III | Maintenance after first-line therapy | Nivolumab +/− Ipilimumab | 834 | OS | No difference in survival |
| **KEYNOTE-028** | Ib | Recurrent/refractory | Pembrolizumab | 24 | Safety, Tolerability and ORR | Main AEs: Asthenia-fatigue-cough ORR 33% |
| **KEYNOTE-158** | II | Recurrent/refractory | Pembrolizumab | 107 | ORR | ORR 18.7%, 35.7% PDL-1$^+$ 6% PDL-1$^-$ |
| **CheckMate-032** | I/II | Refractory/Recurrent | Nivolumab +/− Ipilimumab | 216 | ORR | N3 10%; N1I3 23%; N3I1 19% Promising results |
| **CheckMate-331** | III | Recurrent/Refractory | Nivolumab | 284 receiving nivolumab vs. 285 on chemotherapy | OS | No improvement in OS |

## 5. Biomarkers of Immunotherapy in SCLC

To date, attempts at correlating immune biomarkers to treatment response in SCLC have been disappointing. Although some patients respond dramatically well, others fail to show any benefit, creating controversy and hesitation in the use of immunotherapy as a therapeutic agent in SCLC. Hence, finding biomarkers that lead to the subcategory of patients with SCLC that responds the most to immunotherapy becomes primordial to identify those patients deriving the most benefit, and to attain improved clinical responses.

### 5.1. PD-L1 and PD-L1 Combined Score

PD-L1, an inhibitory checkpoint used by tumor cells to evade immunity, has been a useful marker to predict the benefit of immune checkpoint inhibitors in several malignancies.

In previous studies of SCLC with checkpoint inhibitors, PD-L1 expression seemed to vary between 2% and 83%, most showing less than 50% PD-L1 expression [40]. Disparities were not well understood, but were attributed to differences in staining antibodies, scoring algorithms, tissue biopsy types, and detection platforms [40]. However, the usefulness of PD-L1 expression in SCLC was retrospectively tested in multiple clinical trials where immunotherapy has shown benefit [41].

Notably, substantial PD-L1 expression occurs on stroma cells of SCLC tissue samples, including tumor-infiltrating immune cells and less in tumor cells. For example, in KEYNOTE-028 where only PD-L1 positive tumors were included, PD-L1 positivity definition was set to include membranous PD-L1 expression in ≥1% of tumor and associated inflammatory cells or positive staining in stroma. Only 31.7% of patients tested positive and were subsequently included in the study [35], with the final results showing

an objective response rate (ORR) of 33% in PD-L1 positive refractory ES-SCLC patients receiving pembrolizumab [35]. In KEYNOTE-158, a PD-L1 combined positive score (CPS) defined as the ratio of PD-L1 positive cells to the overall number of tumor cells was also used with positivity defined by a CPS ≥1. Tumors were PD-L1 positive in 39% of patient samples (*n* = 42). Pretreated ES-SCLC patients with a positive score receiving pembrolizumab monotherapy had better responses than PD-L1 negative patients (35.7% versus 6%), with improved survival in favor of PD-L1 positive samples (median overall survival of 14.6 months versus 7.7 months in favor of positive CPS) [36]. The retrospective pooled analysis of patients with recurrent or metastatic disease treated with pembrolizumab from both studies (KEYNOTE-028, KEYNOTE-158) showed that 88% of responsive patients had PD-L1 positive tumors, with a promising objective response rate of 19.3% and a durable anti-tumor activity, implying a possible correlation between PD-L1 positivity and response [42].

On the other hand, in IMpower 133, the trial that led to the approval of atezolizumab in addition to chemotherapy in the first line treatment of ES-SCLC, the benefit was noted across all patient subgroups regardless of PD-L1 status [43]. Exploratory biomarker analysis included PD-L1 expression on both tumor cells as well as tumor-infiltrating immune cells, with end points including the assessment of efficacy based on PD-L1 expression levels as well as blood-based TMB. PD-L1 prevalence was limited and thereby only the exploratory overall survival (OS) was evaluated at ≥1% and ≥5% on tumor or immune cells, whereas the progression free survival (PFS) and the response rates were only assessed based on the ≥1% cutoff. A total of 34% of the intention-to-treat population had PDL-1+ status (*n* = 64 (receiving atezolizumab) +73 (receiving placebo)/total of 403 enrolled), of which 52.6% (*n* = 72) had a ≥1% value, and 21.2% (*n* = 29) ≥5%, with the rest having positive immunohistochemistry but values <1% (*n* = 65, 47.4%). Median overall survival observed in atezolizumab plus carboplatin/etoposide versus placebo plus carboplatin/etoposide among those with PD-L1 expression on tumor cells or tumor-infiltrating immune cells <1%, ≥1%, and ≥5% was 10.2 vs. 8.3 months (HR = 0.51, 95% CI = 0.30–0.89), 9.7 vs. 10.6 months (HR = 0.87, 95% CI = 0.51–1.49), and 21.6 vs. 9.2 months (HR = 0.60, 95% CI = 0.25–1.46). Similarly, progression-free survival (PFS) with atezolizumab plus carboplatin/etoposide versus placebo group was 5.4 vs. 4.2 months (HR = 0.52, 95% CI = 031–0.88) in the PD-L1 < 1% groups, and 5.1 vs. 5.5 months (HR = 0.86, 95% CI 0.51–1.46) in the PD-L1 ≥ 1% group. Notably, response to atezolizumab was independent of PD-L1 expression as well [43].

In CASPIAN trial, of 277 patients with evaluable samples, 151 patients were in the durvalumab group, PD-L1 expression was low with 5% expression of ≥1% in tumor cells (TC) and 22% in immune cells (IC). There was no significant impact of PD-L1 on treatment effect between arms for OS, PFS or ORR [44].

Furthermore, in CheckMate-032, patients with recurrent disease receiving nivolumab after two previous lines of treatment did not show difference in response rates according to PD-L1 status [37].

As such, PD-L1 expression failed to provide consistent correlations with response to immunotherapy in ES-SCLC [13], and alternative biomarkers are being investigated.

### 5.2. Tumor Mutational Burden (TMB) and Micro-Satellite Instability (MSI)

Despite its strong correlation with smoking, and known mutational burden, SCLC is a heterogeneous disease with different genomic profiles. TMB and MSI are two biomarkers that have proven predictive potential in other solid tumors and have been tested in SCLC.

Investigators of CheckMate-032 have studied TMB as a potential predictive biomarker of response to CPIs. TMB was defined as the total number of somatic missense mutations, and patients were divided into groups: low, with 0 to <143 mutations; medium, 143 to 247 mutations, and high with ≥248 mutations. Within both the nivolumab monotherapy and nivolumab plus ipilimumab treatment groups, ORR were higher in those patients with high TMB (21.3% and 46.2%, respectively) than in patients with low (4.8% and

22.2%, respectively) or medium (6.8% and 16.0%, respectively) TMB. One-year progression-free survival rates were higher in the high TMB group (21.2% and 30% for nivolumab monotherapy and nivolumab plus ipilimumab, respectively) compared with the low (not calculable vs. 6.2%, respectively) or medium (3.1% vs. 8.0%, respectively) TMB groups. Similar trends were observed for overall survival [45]. There was a tendency towards a positive correlation, yet more studies are needed, with clear thresholds of detection standardized.

In IMpower133, among 346 patients with available blood-based tumor mutational burden (bTMB) status, OS benefit was noted in both the bTMB high and low subgroups with cut-offs of 10 and 16, respectively. Hazard ratios were 0.73 (95% CI = 0.49–1.08) and 0.73 (95% CI = 0.53–1.00) for bTMB scores <10 and $\geq$10, and 0.79 (95% CI = 0.60–1.04) and 0.58 (95% CI = 0.34–0.99) for scores <16 and $\geq$16 [43]. Similarly, in CASPIAN trial, across three arms of the study, 283 patients (35% of the intention to treat population) were evaluable for tissue based TMB, and it was not predictive of a differential treatment effect for regarding OS, PFS, or ORR [28].

Mismatch repair (MMR) deficiency promotes DNA instability and leads to double-strand breaks, rendering tumor cells prone to the formation of neo-antigens and more susceptible to immune activation. MMR-deficient solid tumors, also called micro-satellite instability (MSI) high tumors, have attained approval for use of immunotherapy. Previous reports from SCLC samples have mentioned a microsatellite instability rate up to 45% [46], with a study from Switzerland mounting up to 76% microsatellite alterations in plasma DNA of SCLC patients [47]. However, in the phase II KEYNOTE-158 that showed benefit of pembrolizumab in this category of patients, there was a very small number of patients with ES-SCLC (four patients), insufficient to label MMR/MSI as a proper immune biomarker in this setting.

### 5.3. SCLC Immunogenicity

Not only does SCLC have a low PD-L1 expression, but certain SCLC phenotypes have shown to have immunosuppressive features affecting the tumor microenvironment. Lower rates of tumor infiltrating lymphocytes were noted with a rather immune-tolerant behavior of immune cells and CD8/CD3 ratios appear to be decreased [17,21]. However, the recent unraveling of specific SCLC categories led to the discovery of SCLC-I, with higher PD-L1 expression, as well as higher expression of other targetable immune checkpoints such as CD80/86, ICOS, TIGIT, TIM3, and LAG3, all of which have been shown to predict improved CPI-induced outcomes [17].

Trying to apply SCLC subtypes to IMpower133, a robust expression of the SCLC-I group was noted in the study population. IMpower-133 patients were stratified according to SCLC variants, and Gay et al. noted a trend to a higher median OS with the use of atezolizumab versus placebo in addition to EP in the SCLC-I subgroup (18 months vs. 10 months) (HR, 0.566; 95% CI, 0.321–0.998) [17]. In contrast, the addition of atezolizumab versus placebo produced more modest gains in median OS in SCLC-P (9.6 versus 6 months), SCLC-A (10.9 vs. 10.6 months), and SCLC-N (10.6 vs. 9.4 months) [17].

A deeper exploration of immune phenotyping of SCLC was through the study of MHC-I expression in SCLC by Mahadaven et al. [48]. Epigenetic regulators responsible for immunologic plasticity such as TAP1 and STING in addition to EZH2 were associated with MHC-I downregulation [49]. Inhibition of MHC-I downregulation by EZH2 inhibitors con-comitantly with STING agonists may lead to up-regulation of immune-activated cytotoxic T-cell response with consequently improved responses to CPI [50].

Perhaps immune phenotyping, and activation of immune profile through epigenetic modulation of SCLC cells is the missing piece of the puzzle, and further consolidative research on the matter is eagerly awaited.

### 5.4. SLFN11

Finally, the protein Schlafen 11 (SLFN11) is a recently recognized biomarker of SCLC response to platinum-based chemotherapy as well as PARP inhibition. It is a mediator in DNA damage and causes an irreversible replication block through a DNA/RNA helicase leading to cell death [51]. Given the role of SLFN11 within the STING pathway and its known effect on immune response [52], SLFN11 as well as DNA damage repair capacity are being studied as predictive biomarkers to immunotherapy [12,53]. An ongoing clinical trial (SWOG 1929) where all patients with ES-SCLC receive induction frontline EP + atezolizumab but are consequently randomized to receive maintenance atezolizumab with or without talazoparib, is prospectively evaluating for SLFN11 positivity [54].

## 6. Alternative Immune Activating Mechanisms

Different alternative immune-activating mechanisms have been attempted in ES-SCLC including but not limited to cytokines and interferons, use of specific vaccines, use of targeted agents, adoptive cell transfer.

### 6.1. Cytokines/Interleukins/Interferons

Given the role inflammatory markers play in immune plasticity and shaping the tumor immune microenvironment, multiple agents were attempted such as IL-2 [55], and interferons α and γ with chemotherapy without any difference in response [56], and only increased risk of adverse events.

### 6.2. Vaccines

With the intention of stimulating targeted immunity against some highly selective antigens expressed on SCLC tumor cells, few vaccine types were attempted.

### 6.2.1. Fucosyl GM1 and Polysialic Acid

Fucosyl GM1 is a monosialoganglioside highly expressed in SCLC. A synthetic version of it was tested as maintenance therapy with a mountable immune response [57]. Similarly, few pre-clinical studies have attempted polysialic acid vaccines with a robust anti-tumoral antibody response, but significant toxicity [58].

### 6.2.2. Dendritic Cell Vaccines

After dendritic cells transduced with adenovirus expressing wild-type p53 were tested in ES-SCLC patients refractory to chemotherapy, with note of regained response to chemotherapy post progression after vaccination [59], NCT03406715 was started to evaluate the combination of these vaccine with ipilimumab and nivolumab in patients with recurrent SCLC [60,61].

### 6.3. Adoptive Cell Therapy

Adoptive cell therapy is the transfer of chimeric antigen receptor T cells (CAR-T) directed against common antigens such as DLL3 highly expressed in SCLC, with AMG 119 being a DLL3 CAR-T cell tested in SCLC [51]; however, the trial is currently suspended, and recent results from ASCO 2021 show remarkable responses with DLL3-targeted bispecific T-cell engager AMG 757 Tarlatamab [60], that is also being tested in combination with PD-1 inhibition (AMG 404) [62].

Other mechanisms of immune activation are also under-study such as GITR [63], TIGIT [64], and OX40 [65] stimulators, alone or in combination with immunotherapy in early phase trials.

## 7. Discussion and Conclusions

SCLC is a very aggressive disease characterized with a high metastatic potential, and early recurrence after treatment. For a long time, despite multiple efforts, there were no changes in the field until the advent of immunotherapy with atezolizumab and durvalumab

in combination with chemotherapy providing improvement in disease response as well as survival outcomes. However, responses have been modest and not durable in all patients. It is empirical to find the right sub-category of patients that will respond best to immunotherapy.

Despite hypotheses of SCLC immunogenicity given the correlation with paraneoplastic diseases as well as the direct causality of smoking that leads to increased DNA damage, and mutational variability, some SCLC cell lines seem to have relative immune tolerance. Previously tested predictive biomarkers such as PD-L1, TMB, and MSI did not seem to be efficient in SCLC. It was only until recently that the variable subtyping of SCLC showed different characteristics and different tumor behaviors with SCLC-I being pro-immunogenic, with the possibility of facilitated determination of this category that seems to be the most responsive to checkpoint inhibition. Furthermore, the elasticity of immune microenvironment and the possibility of immune phenotype switching can potentially be explored therapeutically.

Based on two large, randomized trials showing very similar results, IMpower-133 and CASPIAN, with statistically significant outcome improvements, platinum-based doublet chemotherapy plus etoposide combined to atezolizumab or durvalumab is now the new standard of care approach for ES-SCLC. In fact, CASPIAN trial confirmed the benefit of immunotherapy, replicating the results of IMpower-133 after a few negative immunotherapy studies, and filling the gap of some previously unanswered questions.

Thereby, it is important to highlight the differences between the two studies that shed light on few considerations in clinical practice. One point is the exclusive use of carboplatin in IMpower 133, posing questions on whether the type of platinum used would matter in combination with immunotherapy. Fortunately, this question was answered by CASPIAN that allowed carboplatin or cisplatin to be used as per investigators' choice with no differences observed between the two subgroups. Another consideration is the intensity of standard therapy comparative arm, with IMpower 133 allowing only four cycles of treatment in either arm, whereas the control arm in CASPIAN allowed up to six cycles of chemotherapy, and once again the combination of immunotherapy + chemotherapy proved superiority subsiding doubts of unjust comparisons. Furthermore, IMpower 133 included 10% of patients receiving PCI in both arms, whereas in CASPIAN, only the control arm was permitted to receive optional PCI, with no difference noted in the incidence of brain metastases. However, patients with asymptomatic treated brain metastases performed poorly in all arms of both studies, yet the number of patients with asymptomatic treated brain metastases was small, and no patient with untreated brain metastases was allowed, rendering the question of management of patients with brain metastases an unmet need, and a significant limitation of the two studies [66]. Three other clinical controversies not tackled in IMpower 133, and CASPIAN include the absence of consolidative thoracic radiotherapy, the exclusion of patients with disease-related borderline and/or poor performance status, as well as patients with disease-related paraneoplastic auto-immune disorders. As for radiation therapy, although consolidative radiation therapy has shown to improve survival in patients with good extra-thoracic disease control after first-line chemotherapy [67], in addition to multiple hypotheses regarding the role of radiation therapy in increasing antigen presentation with possibility of an enhanced immune-response, the risk of pneumonitis needs to be evaluated before this is implemented in clinical practice. On the other hand, regarding patients with poor performance excluded from these studies, two ongoing clinical trials are trying to answer this question [68,69].

Finally, despite multiple studies with immunotherapy combinations in SCLC, only IMpower 133 and CASPIAN using the anti-PD-L1 inhibitors atezolizumab and durvalumab respectively proved to be significantly positive. It is unknown whether there is a biologic difference between PD-1 and PD-L1 inhibitors in SCLC or whether this is a coincidence based on the methodologic features of the clinical trials available. More investigation is warranted.

In conclusion, the addition of PD-L1 inhibitors to front-line chemotherapy represents a new therapeutic option in ES-SCLC providing a much-awaited gain for a significant number of patients with this devastating condition and is hopefully the step in the door towards many other immune-modulating options for the future. Multiple resistance mechanisms are investigated, and perhaps targeting those mechanisms can provide options for synergy and enhanced responses.

**Author Contributions:** Both authors R.E.S. and N.B. have contributed equally to the writing and drafting of the manuscript. All authors have read and agreed to the published version of the manuscript.

**Funding:** This article, as well as several others in this Special Issue, was supported by grants from Amgen Canada, AstraZeneca Canada Inc., Eisai Canada Limited, Hoffman La Roche Canada (journal publication fees only), Jazz Pharmaceuticals Canada Inc., Novartis Canada, Sanofi Canada, Pfizer Canada Inc. Funds were used to pay journal publication fees, provide administrative support and honorariums for some authors. These entities did not influence the content of the articles, nor did they review the article prior to publication.

**Conflicts of Interest:** The authors declare no conflict of interest. Authors confirm that all figures and tables are original and are not replicated from any other source.

## Abbreviations

| | |
|---|---|
| SCLC | Small cell lung cancer |
| ES-SCLC | extensive-stage SCLC |
| PE | platinum-etoposide |
| TMB | Tumor Mutational Burden |
| Rb1 | retinoblastoma1 |
| DDR | DNA damage response |
| CHK1 | checkpoint kinase 1 |
| ATR | ataxia telangiectasia and RAD3-related protein |
| ATM | ataxia telangiectasia mutated |
| AURK | aurora kinase |
| EZH2 | enhancer of zeste homology 2 |
| LSD1 | lysine-specific demethylase 1A |
| DLL3 | delta-like ligand-3 |
| SSTR2 | somatostatin receptor 2 |
| TAZ | transcriptional coactivator with PDZ-binding motif |
| PD-L1 | Programmed death ligand-1 |
| BTK | Bruton Tyrosine Kinase |
| BTKi | Bruton Tyrosine Kinase inhibition |
| CPS | PD-L1 combined positive score |
| MSI | Micro-Satellite Instability |
| bTMB | blood-based tumor mutational burden |
| MMR | Mismatch repair |
| SLFN11 | protein Schlafen 11 |
| CPIs | check point inhibitors |
| PCI | prophylactic cranial irradiation |
| ORR | overall response rate |
| mPFS | median progression free survival |
| HR | hazard ration |
| CI | confidence interval |
| mOS | median overall survival |
| OS | overall survival |
| irBORR | best immune related ORR |

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
