# Peer review of "Immunotherapy in Extensive-Stage Small Cell Lung Cancer"

_curroncol, doi:10.3390/curroncol28050347_

Round 1

Reviewer 1 Report

Thank you for the opportunity to review this manuscript on the utilization of immunotherapy for patient with extensive-stage small cell lung cancer. The authors should be commended on a succinct yet comprehensive review of the existing literature, outlining the basic biology and the clinical evidence of ICIs as first-line and second-line treatment modalities. In addition, the authors touched on the limited evidence for prognostic biomarkers. I thoroughly enjoyed reading the manuscript and do not have substantial changes to suggest. As a minor point, I hope the authors can outline some of the limitations of the IMPower133 and CASPIAN studies, e.g. absence of radiotherapy in IMPower. 

Author Response

Thank you for your kind words. We appreciate your review, as well as your most pertinent comments. As suggested, comparison of the 2 trials: IMpower-133 and CASPIAN was performed, with mention of pertinent limitations to be considered for clinical practice: intensity of standard therapy arm, PCI, absence of thoracic radiation therapy, question of brain metastases, exclusion of patients with poor performance, as well as exclusion of patients with disease-related auto-immune manifestations.

“Based on two large, randomized trials showing very similar results, IMpower-133 and CASPIAN, with statistically significant outcome improvements, platinum-based doublet chemotherapy plus etoposide combined to atezolizumab or durvalumab is now the new standard of care approach for ES-SCLC. In fact, CASPIAN trial confirmed the benefit of immunotherapy, replicating the results of IMpower-133 after a few negative immunotherapy studies, and filling the gap of some previously unanswered questions.

Thereby, it is important to highlight the differences between the 2 studies that shed light on few considerations in clinical practice. One point is the exclusive use of carboplatin in IMpower 133, posing questions on whether the type of platinum used would matter in combination with immunotherapy. Fortunately, this question was answered by CASPIAN that allowed carboplatin or cisplatin to be used as per investigator’s choice with no differences observed between the two subgroups. Another consideration is the intensity of standard therapy comparative arm, with IMpower 133 allowing only 4 cycles of treatment in either arm, whereas the control arm in CASPIAN allowed up to 6 cycles of chemotherapy, and once again the combination of immunotherapy + chemotherapy proved superiority subsiding doubts of unjust comparisons. Furthermore, IMpower 133 included 10% of patients receiving PCI in both arms, whereas in CASPIAN, only the control arm was permitted to receive optional PCI, with no difference noted in the incidence of brain metastases. However, patients with asymptomatic treated brain metastases performed poorly in all arms of both studies, yet the number of patients with asymptomatic treated brain metastases was small, and no patient with untreated brain metastases was allowed, rendering the question of management of patients with brain metastases an unmet need, and a significant limitation of the two studies[66]. Three other clinical controversies not tackled in IMpower 133, and CASPIAN include the absence of consolidative thoracic radiotherapy, the exclusion of patients with disease-related borderline and/or poor performance status, as well as patients with disease-related paraneoplastic auto-immune disorders. As for radiation therapy, although consolidative radiation therapy has shown to improve survival in patients with good extra-thoracic disease control after first-line chemotherapy[67], in addition to multiple hypotheses regarding the role of radiation therapy in increasing antigen presentation with possibility of an enhanced immune-response, the risk of pneumonitis needs to be evaluated before this is implemented in clinical practice. On the other hand, regarding patients with poor performance excluded from these studies, 2 ongoing clinical trials are trying to answer this question[68,69].”

Reviewer 2 Report

Nice and comprehensive review.

Only minor comments :
1. wold be nice to add the last FU from the CASPIAN:

LBA61 - Durvalumab ± tremelimumab + platinum-etoposide in first-line extensive-stage SCLC (ES-SCLC): 3-year overall survival update from the phase III CASPIAN study:

https://oncologypro.esmo.org/meeting-resources/esmo-congress-2021/durvalumab-tremelimumab-platinum-etoposide-in-first-line-extensive-stage-sclc-es-sclc-3-year-overall-survival-update-from-the-phase-iii-casp

The latest results for Imfinzi plus chemotherapy showed sustained efficacy after a median follow up of more than three years for censored patients, with a 29% reduction in the risk of death versus chemotherapy alone (based on an HR of 0.71; 95% CI 0.60-0.86; nominal p=0.0003). Updated median OS was 12.9 months versus 10.5 for chemotherapy.

2. In the discussion should be highlited that we have only 2 positive trials - caspian ind imp133- boyth of the agents are PDL1is, while with PD1is negative trials, this might be explained by the differences way of action between the 2 groups.

Author Response

Thank you for your comments and pertinent suggestions.

  • As proposed, addition of the 3-year follow-up discussed at ESMO congress 2021 was added:

“In a recent update of >3-years of follow-up (39.4 months, 86% maturity of data) for censored patients in CASPIAN trial, discussed at the ESMO congress 2021, at the data cut-off date of 27 March 2021, durvalumab addition to chemotherapy showed sustained clinically significant OS benefit in favor of durvalumab with a mOS of 12.9 months versus 10.5 months when compared to chemotherapy alone (HR 0.71, 95% CI 0.6-0.86, p=0.0003). At the 2-year landmark, 22.9% versus 13.9% of patients were alive in favor of durvalumab. At 36 months,17.6% vs 5.8% of pts were alive also in favor of durvalumab. Regarding the durvalumab+ tremelimumab+ chemotherapy arm, OS was numerically improved versus chemotherapy with 15.3% of patients alive at 36 months, however, without any statistically significant difference (HR 0.81, 95% CI 0.67–0.97; p=0.02)[30].”

  • It is quite intriguing to look up the reasons behind the differences in trial where only PD-L1 inhibitors resulted in clinically significant results despite several immunotherapy trials in ES-SCLC. Whether it is a class effect via mechanistic variations (effect on intra-tumoral T-cells, PD-L1 receptor variation, and existence mostly on immune cells), or whether it is solely related to difference in methodology of different studies with possibly sicker patients being included in the other trials; the reason remains ambiguous and requires further investigation. As such, and as per your recommendations, we only brought up that question and mentioned the need for more studies on the matter.

“Finally, despite multiple studies with immunotherapy combinations in SCLC, only IMpower 133 and CASPIAN using the anti-PD-L1 inhibitors atezolizumab and durvalumab respectively proved to be significantly positive. It is unknown whether there is a biologic difference between PD-1 and PD-L1 inhibitors in SCLC or whether this is a coincidence based on the methodologic features of the clinical trials available. More investigation is warranted.”